# Genomic prediction of rice mesocotyl length indicative of directing seeding suitability using a half-sib hybrid population

Liang Chen[1], Jindong Liu[1], Sang He[1]*, Liyong Cao[2]*, Guoyou Ye[1,3]

**1** CAAS-IRRI Joint Laboratory for Genomics-Assisted Germplasm Enhancement, Agricultural Genomics Institute in Shenzhen, Chinese Academy of Agricultural Sciences, Shenzhen, China, **2** Key Laboratory for Zhejiang Super Rice Research, China National Rice Research Institute, Hangzhou, China, **3** Rice Breeding Innovations Platform, International Rice Research Institute, Metro Manila, Philippines

* hesang@caas.cn (SH); caoliyong@caas.cn (LC)

**Data Availability Statement:** All relevant data are within the paper and its Supporting information files.

## Abstract

Direct seeding has been widely adopted as an economical and labor-saving technique in rice production, though problems such as low seedling emergence rate, emergence irregularity and poor lodging resistance are existing. These problems are currently partially overcome by increasing seeding rate, however it is not acceptable for hybrid rice due to the high seed cost. Improving direct seeding by breeding is seen as the ultimate solution to these problems. For hybrid breeding, identifying superior hybrids among a massive number of hybrids from crossings between male and female parental populations by phenotypic evaluation is tedious and costly. Contrastingly, genomic selection/prediction (GS/GP) could efficiently detect the superior hybrids capitalizing on genomic data, which holds a great potential in plant hybrids breeding. In this study, we utilized 402 rice inbred varieties and 401 hybrids to investigate the effectiveness of GS on rice mesocotyl length, a representative indicative trait of direct seeding suitability. Several GP methods and training set designs were studied to seek the optimal scenario of hybrid prediction. It was shown that using half-sib hybrids as training set with the phenotypes of all parental lines being fitted as a covariate could optimally predict mesocotyl length. Partitioning the molecular markers into trait-associated and -unassociated groups based on genome-wide association study using all parental lines and hybrids could further improve the prediction accuracy. This study indicates that GS could be an effective and efficient method for hybrid breeding for rice direct seeding.

## 1. Introduction

Rice, as an essential food crop, feeds more than half of the world human population. To meet this huge demand, modern and advanced agricultural technologies were used to improve rice production. Mechanized direct seeding can conspicuously improve the planting efficiency, which has been widely adopted in rice production [1,2]. However, direct seeding in rice also faces some difficulties such as the low emergence rate, irregular emergence and easy lodging of seedlings [3]. Increasing seeding rate might resolve the problems for inbred lines yet it is not

**Funding:** This work was supported by National Key R&D Program of China (2020YFE0202300), the Young Elite Scientists Sponsorship Program by CAST (YESS, 2020QNRC001) and the Agricultural Science and Technology Innovation Program (ASTIP). The funders had no role in study design, data collection and analysis, decision to publish, or preparation of the manuscript.

**Competing interests:** The authors have declared that no competing interests exist.

an option for hybrids due to the high seed cost. Considering the advantage of exploiting heterosis from hybrids in rice breeding, e.g., Jumin et al. [4] reported that the $F_1$ hybrids had an approximate 20% higher grain yield than the inbred lines, developing hybrid varieties suited for direct seeding is of great significance and now has become the focus of many rice breeding programs. Several traits that are indicative of the ease of direct seeding have been identified. One representative is the mesocotyl length as a long mesocotyl could markedly improve emergence rate, early vigor and lodging tolerance [5]. However, modern varieties developed for well irrigated ecosystem by transplanting regularly normally have short mesocotyl ($\leq$1.0 cm) [6]. Thereby, it is crucial to breed hybrid varieties with long mesocotyl for direct seeding.

For hybrid development, identifying excellent hybrid combinations is pivotal. Since the number of hybrids producible is far more than the number of parental lines, selecting the exceptional combinations to be produced and tested is difficult for breeders. Accurately predicting hybrid performance so that only promising combinations are field-tested has long been a research hotspot. Mid-parent performance, general and specific combining abilities, genetic distance between parental lines estimated using traits or markers have been tested but are of limited usefulness depending on the traits and parental populations [7–10]. Currently, genomic selection (GS) has been widely used to predict hybrid performance in various crops. In GS, the performances of untested genotyped plant individuals are predicted based on the genomic relationship between them and a well-composed training set with both phenotypic and genotypic data. Riedelsheimer et al. [11] used 285 maize inbred lines to test cross with two maize varieties to obtain 570 hybrids. Through the cross-validation within the hybrids, it was found that the prediction accuracies of seven traits ranged from 0.72 to 0.81 with the heritabilities varied from 0.82 to 0.98. Xu et al. [12] predicted the yield of all possible 21,945 hybrid progenies using 278 hybrids generated from random crossings of 210 recombinant inbred lines. If the top 10 hybrid combinations were selected for hybrid breeding, the yield would be increased by 16%. The inclusion of non-additive effects, i.e., dominant and epistatic effects, in genomic prediction brought no benefit in real data but showed usefulness in simulation when non-additive effects were simulated [12]. Thereby, it is potentially profitable to accommodate the non-additive effects though the additive effects are dominant [12]. In addition to genomic information, other omics information is also able to assist genomic prediction. Xu et al. [13] reported that combining the parental phenotypes with other predictors can significantly improve the predictability of yield-related traits in rice. Fu et al. [14] used four methods including multiple linear regression, PLS, SVM and transcriptome distance to predict the phenotype of maize hybrids and found that the prediction based on transcriptome distance was the most accurate. Xu et al. [15] used the metabolic data of 210 inbred lines to predict the yield of their hybrids and found that the prediction ability was almost twice than that of genome markers. Westhues et al. [16] revealed the advantages of combining transcriptome data with genomic data measured for parents for the prediction of untested hybrids. In the prediction of hybrid rice, Wang et al. [17] compared the predictability of combinations between multi-omics data including genomics, transcriptome and metabolome data and eight GS methods, finding that the GBLUP approach integrating genomics and metabolome data performed overall the best.

The abovementioned studies have shown that GS holds the potential to effectively predict yield and yield-related traits in hybrid rice, but no study ever investigated the potential of GS on hybrid rice mesocotyl length which is indicative to direct seeding. In this study, we measured the mesocotyl length of 402 rice inbred lines including a famous male sterile line Taifeng A and their 401 hybrid progenies produced by test-crossing the 401 lines with Taifeng A as the female parent. We examined several genomic prediction scenarios including mid-parental value prediction, marker-assisted selection (MAS) and genome-wide association

study (GWAS). Our major aim is to find the optimal hybrid rice prediction scenario with the highest prediction accuracy of mesocotyl length to disclose the potential of using genomic selection to accelerate the breeding of hybrid rice suited to direct seeding.

## 2. Materials and methods

### 2.1 Rice materials

The 402 rice varieties used in this study are mainly from South Asia, Southeast Asia and South China, conserved in the International Rice Research Institute. The specific variety information was shown in S1 Table. The 401 F1 hybrid populations were produced by test crossing 401 rice varieties (as male parent) with a widely used male sterile line Taifeng A (as female parent).

### 2.2 Phenotypic data

A randomized complete block design with three replicates was used to layout the test of mesocotyl length measurement for all parental lines and hybrids. In order to minimize the impact of environment on the phenotypic performances of hybrid and its parent, each hybrid was planted next to its male parent. Total 15 full seeds per variety were taken for sowing at the depth of 6 cm in each block. The plastic cavity tray with 50 hole was used for sowing. The hole depth, upper diameter and bottom diameter of the tray was 9.5 cm, 4.5 cm, and 2.1 cm, respectively. After sowing, each plastic cavity tray was placed in the corresponding plastic pallet with nutrient soil covered the bottom at the depth of 3 cm, and then all the pallets were transported into a large-volume oven to culture at 30 °C under the dark. Keep the soil in the tray and pallet moist until the seeds germinated unearthed. Record the emergence rate every day until that of all varieties reached 100%. After that, take out all seedlings in the hole and wash them with clean water and then randomly select 10 seedlings per variety with uniform rise to take photos. The mesocotyl length measurement was performed using image J (https://imagej.en.softonic.com/). The phenotypic values were adjusted to derive the best linear unbiased estimates (BLUE) using formula: $y = Xb + Zu + e$, where $y$ is the observed phenotypic values of mesocotyl length for all lines and hybrids, $b$ is the block effect, $u$ is the genetic effect, $X$ and $Z$ are the design matrices for $b$ and $u$, $e \sim N(0, I\sigma_e^2)$ is the random residual where $I$ is identity matrix and $\sigma_e^2$ is the residual variance component. Both $b$ and $u$ were regarded as fixed effect. The phenotypic adjustment process was implemented in R [18] using package sommer [19].

### 2.3 Heterosis analysis

The heterosis performance (Hp) was calculated using the formula: Hp = 2 × (F$_1$ − MP) / | P1 − P2 |, where F$_1$ is the performance of the hybrid, MP is the average performance of the two parents, P1 and P2 is the performance of male parent and female parent, respectively. According to the value of Hp, high-parent heterosis (HPH), the mid-parent heterosis (MPH), low-parent heterosis (LPH), and hybrid inferiority (HI) were defined as Hp > 1, 0 < Hp ≤ 1, -1 ≤ Hp < 0, and Hp < -1, respectively [20].

### 2.4 Genomic data

The DNA of 402 inbred rice samples was extracted by CTAB method. The sequencing platform used Illumina Hiseq 2000 (PE 150) (https://www.berrygenomics.com/) from Beirui Gene Company with the sequencing depth 50×. The sequencing reads were against with Japonica reference genome (IRGSP-1.0) (http://rice.plantbiology.msu.edu/index.shtml) by BWA-MEM V0.7.10 (http://bio-bwa.sourceforge.net/bwa.shtml). Repeated reads were classified using the Picard tool (http://broadinstitute.github.io/picard/). The variation sites such as high quality

SNP and INDEL per variety were captured utilizing GATK V3.2.2 (https://gatk.broadinstitute.org/hc/en-us) with the parameter setting of QUAL < 30.0, QD < 10.0, FS > 200.0, MQ Rank Sum < -12.5 and Read Pos Rank Sum < -8.0.

A total of 7,882,841 bi-allelic SNPs were identified for the 402 lines. Quality control for the SNPs followed criteria that 1) remove SNPs with minor allele frequency (MAF) less than 0.05; 2) remove SNPs with genotyping call rate less than 90%; 3) exclude SNPs with heterozygotes rate more than 10%. As a result, 196,640 high quality SNPs retained. Genotype imputation was implemented to impute the missing genotypic profiles of the 196,640 high quality SNPs by IMPUTE2 software [21]. The heterozygotes were all arbitrarily set to missing values and imputed. Once imputation was done, a quality control for linkage disequilibrium (LD) between SNPs was applied to keep independent SNPs. The software PLINK [22] was used with the parameters window size, shifting step, and $r^2$ threshold respectively being set to 50 SNPs, 5 SNPs, and 0.1. Finally, 10,547 independent SNPs were available for the 402 lines. The genotypic data of the 402 lines was provided in S2 Table.

The genotypes of the hybrids were deduced from the genotypes of their parents. Specifically, for a particular SNP, the two types of homozygotes in the parental lines were numerically coded as 0 or 2, indicating the number of copies of the alternative allele. The profile of hybrids was the mean value of the genotypes of their parents, i.e., 0 or 1 or 2.

To investigate the population structure underlying the lines, a cluster analysis based on the SNP genotypic data was performed.

## 2.5 Mid-parental value prediction

The mid-parental values of the hybrids in the test sets of each cross-validation scenario (details can be found in section 2.8) were used as the phenotypically predicted genetic values of the hybrids.

## 2.6 Marker-assisted selection

The marker-assisted selection includes two steps. In the first step, the GWAS was performed in the training set of each cross-validation scenario using a mixed linear model:
$y_r = 1_r\mu + \Sigma_{k=1}^3 PC_k + X_{r_j}b_j + Z_r g_r + e$, where $y_r$ is a r-dimensional vector of adjusted phenotypic values of Mesocotyl length, r is the number of genotypes in the training set, $1_r$ is a r-dimensional vector of ones, $\mu$ is the intercept, $PC_k$ is the k[th] the principal component vector derived from the genomic data, $b$ is the additive genetic effect of j[th] SNP, $X_{r_j}$ is a r-dimensional vector containing genotypic profiles of j[th] SNP, $g_r$ is a r-dimensional vector of additive genetic effects of genotypes following $g_r \sim N\left(0, A_r\sigma_{a_r}^2\right)$, $A_r$ is a r×r-dimensional additive genomic relationship matrix estimated following Yang et al. [23], $\sigma_{a_r}^2$ is the corresponding variance component, $Z_r$ is the design matrix of $g_r$, and $e$ are the random residuals following $e \sim N\left(0, I\sigma_e^2\right)$ where $I$ is identity matrix and $\sigma_e^2$ is the residual variance component. The thresholds of filtering significant SNPs ranged from $5\times10^{-5}$ to 0.01. Once GWAS was done, for each significance threshold, a linear model using the identified trait-associated SNPs (TA-SNPs) was fitted as: $y_r = 1_r\mu + \sum_{j=1}^m X_j b_j + e$ in cross-validation scenarios 1–5 (details can be found in section 2.8), where m is the number of TA-SNPs. The estimated effects of the TA-SNPs $\hat{b}_j$ were accordingly derived. The effective number of TA-SNPs were calculated following Jiang et al. [24]. Briefly, a principal component analysis was performed using the genotypic profiles of all the TA-SNPs. The number of principal components in total explaining 95% variation is the effective number of TA-SNPs. In the second step, the phenotypic values of hybrids in the

test set $\widehat{\boldsymbol{y}_s}$ were predicted using the formula $\widehat{\boldsymbol{y}_s} = \boldsymbol{1}_s\widehat{\mu} + \sum_{j=1}^{m} \boldsymbol{X}_{s_j}\widehat{b_j}$ in cross-validation scenario 1–5, where $\boldsymbol{1}_s$ is a s-dimensional vector of ones, s is the number of hybrids in the test set, $\widehat{\mu}$, $\widehat{\beta}$ and $\widehat{b_j}$ are respectively the estimated values of intercept, and effect of $j^{\text{th}}$ TA-SNP from the linear model in the first step, $\boldsymbol{X}_{s_j}$ is a s-dimensional vector of genotypic profiles of $j^{\text{th}}$ SNP in the test set.

The GWAS analyses were implemented in GCTA software [25] using option "—mlma". The calibration and prediction linear models were fitted in R [18].

## 2.7 Genomic prediction methods

Two BLUP models, GBLUP and EGBLUP, and two Bayesian approaches, BayesB and BayesR were used in genomic prediction. The two BLUP models could be uniformly formulated as $\boldsymbol{y} = \boldsymbol{1}_n\mu + \boldsymbol{Zg} + \boldsymbol{\varepsilon}$ in cross-validation scenarios 1–5 and $\boldsymbol{y_h} = \boldsymbol{1}_h\mu_h + \boldsymbol{W\beta} + \boldsymbol{Z_hg_h} + \boldsymbol{\varepsilon_h}$ in the cross-validation scenario incorporating mid-parental value as a covariate (details can be found in section 2.8), where $\boldsymbol{y}$ is a n-dimensional vector of adjusted phenotypic values of mesocotyl length, n is the number of genotypes in both training and test sets, $\boldsymbol{y_h}$ is a h-dimensional vector of adjusted phenotypic values of mesocotyl length for all hybrids, h is the number of all hybrids, $\boldsymbol{1}_n$ and $\boldsymbol{1}_h$ are n- and h-dimensional vectors of ones, $\mu$ and $\mu_h$ are the intercepts, $\boldsymbol{W}$ is a h-dimensional covariate vector of mid-parental values, $\beta$ is the covariate effect, $\boldsymbol{g}$ and $\boldsymbol{g_h}$ are the n- and h-dimensional vectors of genetic effect of genotyped individuals and genotyped hybrids, $\boldsymbol{Z}$ and $\boldsymbol{Z_h}$ are the design matrices for $\boldsymbol{g}$ and $\boldsymbol{g_h}$, $\boldsymbol{\varepsilon}$ and $\boldsymbol{\varepsilon_h}$ are the random residuals following $\boldsymbol{\varepsilon} \sim N\left(\boldsymbol{0}, \boldsymbol{I}_n\sigma_{\varepsilon}^2\right)$ and $\boldsymbol{\varepsilon_h} \sim N\left(\boldsymbol{0}, \boldsymbol{I}_h\sigma_{\varepsilon_h}^2\right)$ where $\boldsymbol{I}_n$ and $\boldsymbol{I}_h$ are identity matrices, $\sigma_{\varepsilon}^2$ and $\sigma_{\varepsilon_h}^2$ are the variance component of residuals. For GBLUP, the genetic effect $\boldsymbol{g}$ and $\boldsymbol{g_h}$ were the additive effect following $\boldsymbol{g} = \boldsymbol{a} \sim N\left(\boldsymbol{0}, \boldsymbol{A}\sigma_a^2\right)$ and $\boldsymbol{g_h} = \boldsymbol{a_h} \sim N\left(\boldsymbol{0}, \boldsymbol{A_h}\sigma_{a_h}^2\right)$ where $\boldsymbol{a}$ and $\boldsymbol{a_h}$ are an n- and h- dimensional vector of additive genetic effects of genotyped individuals and genotyped hybrids, $\sigma_a^2$ and $\sigma_{a_h}^2$ are corresponding variance components, and $\boldsymbol{A}$ and $\boldsymbol{A_h}$ are the additive genomic relationship matrices [26]. For EGBLUP, the genetic effect $\boldsymbol{g}$ and $\boldsymbol{g_h}$ contain both additive and additive-by-additive epistatic effects assuming $\boldsymbol{g} = \begin{pmatrix} \boldsymbol{a} \\ \boldsymbol{p} \end{pmatrix} \sim$

$MN(\begin{pmatrix} \boldsymbol{0} \\ \boldsymbol{0} \end{pmatrix}, \begin{pmatrix} \boldsymbol{A}\sigma_a^2 & \boldsymbol{0} \\ \boldsymbol{0} & \boldsymbol{A}\#\boldsymbol{A}\sigma_p^2 \end{pmatrix}))$ and $\begin{pmatrix} \boldsymbol{a_h} \\ \boldsymbol{p_h} \end{pmatrix} \sim MN(\begin{pmatrix} \boldsymbol{0} \\ \boldsymbol{0} \end{pmatrix}, \begin{pmatrix} \boldsymbol{A_h}\sigma_{a_h}^2 & \boldsymbol{0} \\ \boldsymbol{0} & \boldsymbol{A_h}\#\boldsymbol{A_h}\sigma_{p_h}^2 \end{pmatrix})$ where #

denotes the Hadamard product, $\boldsymbol{p}$ and $\boldsymbol{p_h}$ are an n- and h-dimensional vector of epistatic genetic effect of genotyped individuals and genotyped hybrids, $\sigma_p^2$ and $\sigma_{p_h}^2$ are corresponding variance components. The genomic heritability was calculated based on the GBLUP model using the formula $h_g^2 = \frac{\sigma_a^2}{\sigma_a^2 + \sigma_{\varepsilon}^2}$ where the variance components were estimated respectively in the populations of lines and hybrids. The two Bayesian approaches could be uniformly formulated as $\boldsymbol{y_r} = \boldsymbol{1}_r\mu + \boldsymbol{X_r\gamma} + \boldsymbol{e}$ in cross-validation scenarios 1–5 and $\boldsymbol{y}_{h_r} = \boldsymbol{1}_{h_r}\mu_h + \boldsymbol{W}_{h_r}\beta + \boldsymbol{X}_{h_r}\gamma_h + \boldsymbol{e_h}$ in the cross-validation scenario incorporated mid-parental value as a covariate, where $\boldsymbol{y_r}$ is a r-dimensional vector of adjusted phenotypic values of mesocotyl length, r is the number of genotypes in the training set, $\boldsymbol{y}_{h_r}$ is a $h_r$-dimensional vector of adjusted phenotypic values of msocotyl length, $h_r$ is the number of genotypes in the training set of hybrids, $\boldsymbol{1}_r$ and $\boldsymbol{1}_{h_r}$ are r- and $h_r$-dimensional vectors of ones, $\mu$ and $\mu_h$ are the intercepts, $\gamma$ and $\gamma_h$ are m-dimensional vector of additive genetic effects of each SNP respectively predicted in the training set and training set of hybrids, m is the number of SNPs, $\boldsymbol{X_r}$ and $\boldsymbol{X}_{h_r}$ are an r × m- and $h_r$ × m-dimensional matrix with elements 0, 1, and 2 representing the copies of alternative alleles of SNPs,

$W_r$ is a $h_r$-dimensional covariate vector of mid-parental values, $e$ and $e_h$ are the random residuals following $e \sim N\left(\mathbf{0}, I_r\sigma_e^2\right)$ and $\varepsilon_h \sim N\left(\mathbf{0}, I_{h_r}\sigma_{e_h}^2\right)$ where $I_r$ and $I_{h_r}$ are identity matrices, $\sigma_e^2$ and $\sigma_{e_h}^2$ are the variance component of residuals. In BayesB, the prior distribution of marker effect is assumed to be a mixture of a $t$ distribution with a fixed probability $\pi$ and a point mass at zero with a probability $1-\pi$ [27]. In BayesR, the marker effect is assumed to follow a mixture of four normal distributions with zero mean and varied variances. The sum of proportions of each normal distribution $\pi = (\pi_1, \pi_2, \pi_3, \pi_4)$ is constrained to unity [28]. The phenotypic values of hybrids in the test set $\widehat{y}_s$ were predicted using the formula $\widehat{y}_s = \mathbf{1}_s\widehat{\mu} + X_s\widehat{\gamma}$ in cross-validation scenarios 1–5 and $\widehat{y}_s = \mathbf{1}_s\widehat{\mu_h} + W_s\widehat{\beta} + X_s\widehat{\gamma_h}$ in the cross-validation scenario incorporating mid-parental value as a covariate, where $\mathbf{1}_s$ is a s-dimensional vector of ones, s is the number of hybrids in the test set, $W_s$ is a s-dimensional covariate vector of mid-parental values, $X_s$ is an s × m-dimensional matrix of genotypic profiles of hybrids in the test set, $\widehat{\mu}$, $\widehat{\mu_h}$ and $\widehat{\beta}$ are respectively the estimated values of intercepts and covariate, and $\widehat{\gamma}$ and $\widehat{\gamma_h}$ are m-dimensional vector of SNP effects respectively predicted in the training set and training set of hybrids.

In different cross-validation scenarios, theoretically the calibration models using hybrids (scenarios 2–5 and the scenario incorporating mid-parental value as a covariate) could predict both additive and dominant genetic effects and part them. However, as only one female was used in our study, deductively, the additive genotypic profiles of the hybrids were completely collinear with their dominant genotypic profiles. Therefore, the additive and dominant genetic effects could not be *de facto* partitioned. Despite this, the collinearity resulted the dominant effect compounded with the additive effect, thereupon the genetic merit of dominant effect was still involved and utilized in the scenarios using hybrids (scenarios 2–5).

The BLUP models were realized in R [18] using package BGLR [29]. The Bayesian approaches were fitted using GCTB software [30]. The iteration times, burn-in, and thinning of all models were set to 30,000, 5,000, and 5 respectively.

## 2.8 Cross-validation scenarios

The 401 hybrids were stochastically and evenly divided into five folds. One fold formed the test set. Five scenarios to compose the training set were considered as follows: Scenario 1) reference hybrids' parents: the parental lines of the hybrids not in the test set formed the training set; Scenario 2) reference hybrids: other four folds of hybrids beside the test set constituted the training set; Scenario 3) reference hybrids and their parents: other four folds of hybrids beside the test set and their parents collectively comprised the training set; Scenario 4) reference hybrids and all lines: other four folds of hybrids beside the test set and all lines were combined as the training set; Scenario 5) reference hybrids and parents of test set: other four folds of hybrids beside the test set and the parental lines of the test set collectively comprised the training set. Another scenario using mid-parental values of all hybrids as a covariate in the prediction models was considered. The training set was constituted by the four folds of hybrids beside the test set. In this scenario, the phenotypic data of all parents and reference hybrids was taken advantage, which contained comparable reference information as scenario 5. Each cross-validation scenario was repeated 20 times, yielding in total 100 times random partitioning of training and test sets for each scenario. The prediction accuracy of GS and MAS was evaluated on the basis of combining five test sets in each repeat of cross-validation. Specifically, the genomic predicted genetic values of the five tests in each repeat of cross-validation were combined and the Pearson correlation coefficient between the combined predicted values and the corresponding adjusted phenotypic values was calculated to measure the genomic prediction accuracy. Thereupon, 20 prediction accuracies from the 20 repeats of cross-validations

were shown for each training set composition scenario. In the mid-parental value prediction scenario, there was no model-training and the predicted genetic values of hybrids in the test set were perpetually the mid-parental values of their parents. Therefore, when the five hybrid test sets in each repeat of cross-validation were combined to measure the prediction accuracy, the predicted values in the combination were invariable disregarding to the samples of training and test sets, that is, the mid-parental values of the total hybrid population. Due to this, there was just one prediction accuracy value in the scenario of the mid-parental value prediction. The scenarios of different training set compositions were illustrated in S1 Fig. All prediction accuracies of GS and MAS were z-transformed for statistical test and analysis of variance (ANOVA).

To investigate the impact of training set size on genomic prediction accuracy, 5% to 80% of reference hybrids in each training set composition scenario were randomly sampled to establish training set subsets with different sizes. The sampling of training set subsets was repeated 20 times for each sampling ratio, yielding in total 2000 times (20×100) calibrations and predictions in marker-assisted selection and genomic prediction.

### 2.9 Classification of SNPs in genomic prediction

The SNP markers were classified into two groups by GWAS. One group consisted of the TA-SNPs identified in GWAS and another group was the remaining genome-wide SNPs. GWAS was implemented respectively using all lines and the total population including all lines and hybrids. The GWAS model was identical to that used in MAS. The threshold of significance determining the TA-SNPs was decided by the best performing MAS model with overall highest prediction accuracy. The GBLUP model was used to validate the effectiveness of classifying markers in genomic prediction. The group of TA-SNPs was respectively fitted as fix and random effect in the model. As fix effect, considering the number of TA-SNPs would be large, a principal component analysis was utilized. The principal components accounting for 95% variation were used in place of TA-SNPs in the model. When TA-SNPs were fitted as random effect, two separate kernels respectively composed by TA-SNPs and remaining genome-wide SNPs were fitted in the GBLUP model. The GBLUP model was implemented in R package BGLR [29] with 30,000 iterations, 5,000 times burn-in, and thinning of 5.

## 3. Results

### 3.1 Phenotypic analysis statistics and population diversity

Results of the phenotypic analysis were summarized in Table 1 and in details shown in S3 Table. The BLUE of mesocotyl length ranged from -0.14 to 5.87 for parent lines and -0.1 to 5.61 for hybrids. Heterosis analysis indicated that among 401 hybrids, 41% show high parent heterosis, 19% shown mid-parent heterosis, 26% shown low parent heterosis and the remaining 14% shown hybrid inferiority (S4 Table and S2 Fig). High parent heterosis was the major contributor to mesocotyl length. The additive effect variance component was 1.22 for parent lines and 1.7 for hybrids. The heritability estimate was 0.8 for parent lines and 0.58 for hybrids.

**Table 1. Range and coefficient of variation (CV) of best linear unbiased estimates (BLUE) of genetic effect of genotypes, and variance components of additive genetic effect ($\sigma_a^2$) and random residual ($\sigma_\varepsilon^2$), and genomic heritability ($h_g^2$) of mesocotyl length, separately estimated from the parental lines and hybrids populations.**

| Population | Size | Range | CV | $\sigma_a^2$ | $\sigma_\varepsilon^2$ | $h_g^2$ |
|---|---|---|---|---|---|---|
| Lines | 402 | [-0.14, 5.87] | 0.74 | 1.22 | 0.30 | 0.80 |
| Hybrids | 401 | [-0.10, 5.61] | 0.78 | 1.70 | 1.21 | 0.58 |

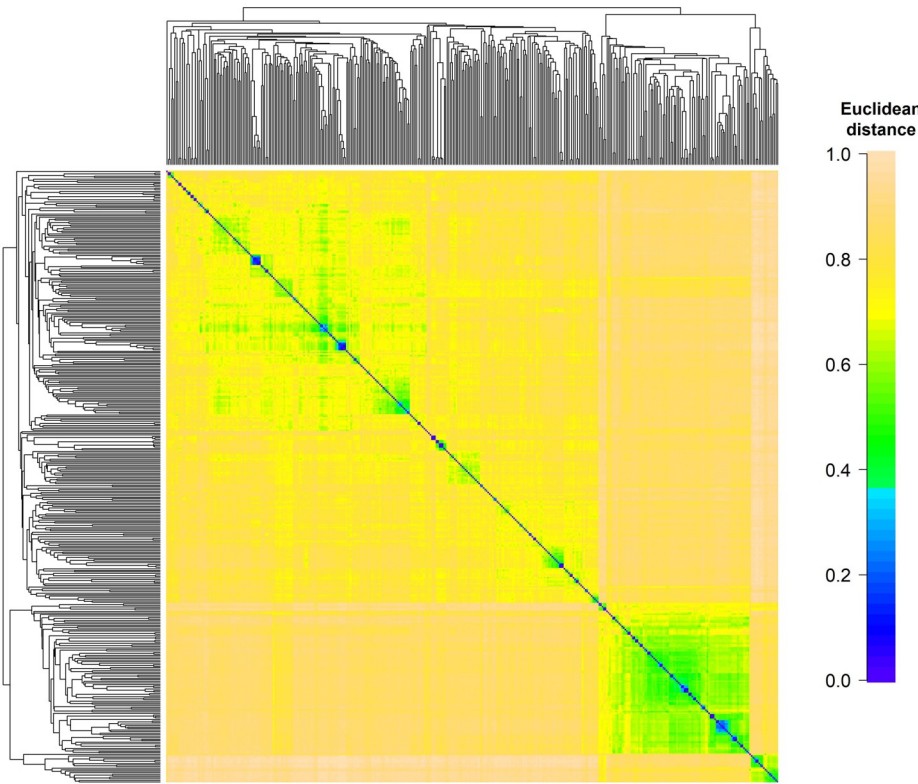

**Fig 1. Pairwise genetic dissimilarities between 401 parental lines based on Euclidean distance.** The average clustering method was used to order the lines.

The genetic diversity of the parental lines was overall high, as indicated by the wider range of genetic similarities between parental lines (Fig 1).

## 3.2 Predictability of marker-assisted selection

The prediction accuracies of MAS in different training set composition scenarios were shown in Fig 2 and S5 Table. Using mid-parental values to predict the performances of hybrids (mid-parental value prediction) resulted in an accuracy of 0.59, which was used as a reference to other prediction scenarios and marked using a red dash line in Fig 2. When the training population contains parental lines of the reference hybrids for cross-validation only (scenario 1), which assumed no data on hybrids is available, the prediction accuracy ranged from 0 to 0.39, which increased to the maximum value with the increase of different significance thresholds (P value) to 0.001, which was obviously lower than the result of mid-parental value prediction. The effect of P value on MAS prediction was significant. When only the reference hybrids were used in the training set (scenario 2), which was the typical scheme commonly applied in other GS studies of rice hybrid prediction, the prediction accuracy ranged from 0 to 0.48, increased to the maximum with the increase of P value to 0.0025, which was lower than the result of mid-parental value prediction. Surprisingly, when P value was 0.0075 or 0.01, the prediction accuracy dramatically dropped to less than 0.15. This might indicate that the increase of number of markers due to a liberal P value brings more noise than signal into the multiple linear regression models we applied. When the reference hybrids and their parental lines were used in the training set (scenario 3), which modelled the situation that phenotypic test was

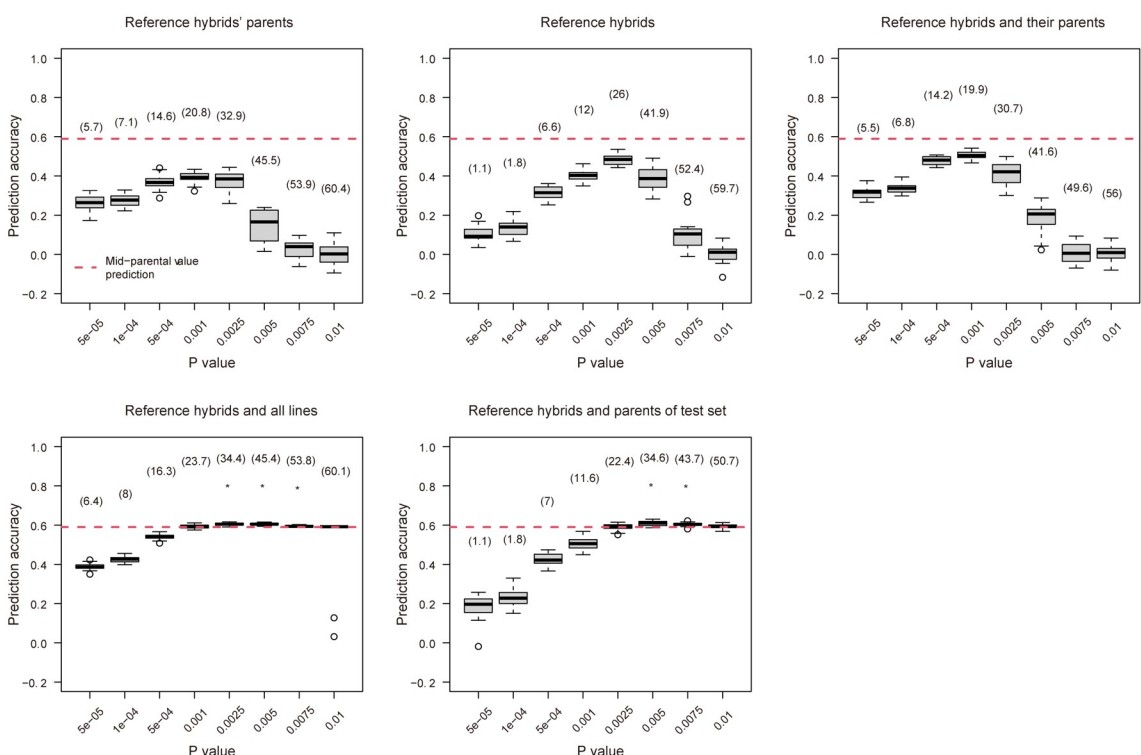

**Fig 2. Prediction accuracies of mesocotyl length using marker-assisted selection with different significance thresholds (P value) for selecting trait-associated SNP markers based on different training set composition scenarios.** The red dash line indicates the prediction accuracy using mid-parental values. The numbers in parentheses indicate the number of effective trait-associated SNPs (TA-SNPs). The asterisks indicate the prediction accuracies based on varying training set composition scenarios were significantly higher ($p < 0.05$, $t$-test) than the prediction accuracy using mid-parental values. All prediction accuracies were z-transformed for statistical test.

conducted for some of the hybrids and their parental lines, the prediction accuracy varied from 0.01 to 0.51 with the maximum value being achieved when P value was 0.001, which was lower than the result of mid-parental value prediction. When the training set contained reference hybrids and their parental lines, and parental lines of the untested hybrids (scenario 4), which modelled the situation that parental lines of untested hybrids have been tested, the prediction accuracy ranged from 0.39 to 0.61, increased to the maximum value with the increase of P value to 0.005. When P value was 0.0025, 0.005 or 0.0075, the prediction accuracies in scenario 4 were significantly higher than the result of mid-parental value prediction. When the training set contained reference hybrids and parental lines of the test set (scenario 5), which assumed the paternal lines of reference hybrids are not helpful to the prediction of test hybrids due to the genetic distance, the prediction accuracy ranged from 0.18 to 0.61, increased to the maximum value with the increase of P value to 0.005. When P value was 0.005 or 0.0075, the prediction accuracy in scenarios 5 was also significantly higher than the result of mid-parental value prediction. Scenario 4 achieved the higher prediction accuracy in a wider range of P values, which was the best scenario.

Two-factor variance analysis showed that the mean variance of the scenario was over three-fold higher than that of the P value, indicating that the training set composition had much higher impact on prediction accuracy than the P value (S6 Table). The interaction between the scenario and P value was also significant but relatively less important (S6 Table). Although

there was no single best P value for all training set compositions, 0.0025 was a better choice when all compositions were considered (Fig 2).

### 3.3 Predictability of genomic prediction

The average prediction accuracies were obtained using different GP models and the result were shown in Fig 3. In the model by GBLUP, the prediction accuracies of scenario 1 to 5 were 0.54, 0.63, 0.6, 0.63 and 0.67, respectively, which were mostly significantly higher than the result of mid-parent value prediction except scenario 1. The prediction accuracy of scenario 2 was significantly higher than that of scenario 1, indicating reference hybrids as training set performed more outstandingly than the parents of reference hybrids. The prediction accuracy of scenario 3 was significantly higher than that of scenario 1, but significantly lower than that of scenario 2, implying combining the parents of reference hybrids with reference hybrids as training set was better than only using parents as training set but inferior to using reference hybrids as training set. However, the prediction accuracy of scenario 4, which was the best group performing in MAS, was significantly higher than that of scenario 3 and equal to that of scenario 2, demonstrating that integrating all parents into the training set consisting of reference hybrids would only marginally improve the predictability. The prediction accuracy of scenario 5 performed the best among all scenarios, which indicated integrating parents of test set into the training set constituted by reference hybrids would significantly improve the predictability in GS.

Comparing different genomic prediction models, their prediction accuracies were quite similar in each scenario except for scenario 4 in which the EGBLUP method performed conspicuously better than other approaches (Fig 3).

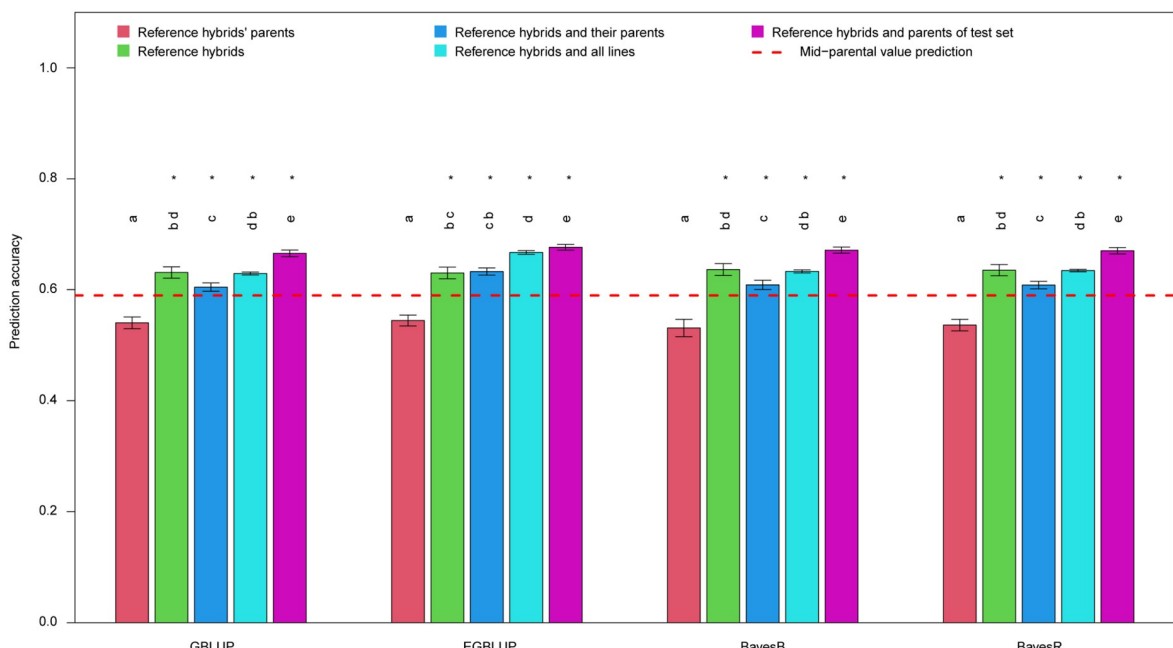

**Fig 3. Genomic prediction accuracies of mesocotyl length using four prediction models based on different training set composition scenarios.** The red dash line indicates the prediction accuracy using mid-parental values. The asterisks indicate the genomic prediction accuracies based on varying training set composition scenarios were significantly higher ($p < 0.05$, $t$-test) than the prediction accuracy using mid-parental values. Different letters above the bars indicated the genomic prediction accuracies of varying scenarios within a specific model were significantly different ($p < 0.05$, $t$-test). All prediction accuracies were z-transformed for statistical test.

In variance analysis, the mean variance of the scenario was over 30 folds higher than that of the prediction model (S7 Table), indicating that the training set composition had much higher effect on prediction accuracy than the prediction model. The interaction between the scenario and the prediction model was also significant but relatively less important (S7 Table).

### 3.4 Using parental performance as covariates in genomic prediction

Previous studies have concluded that incorporating parent information into the model could improve the prediction accuracy [13]. We also used the mid-parental value as a covariate incorporated into the model. Surprisingly, the prediction accuracy was markedly and significantly higher than that of scenario 2 (only using reference hybrids as training set), demonstrating the huge advantage of integrating the mid-parent value as a covariate into the model (Fig 4).

### 3.5 Different training set sizes with subsets of reference hybrids

Next, we investigated the effect of training set size on genomic prediction under different training set compositions. For scenario 2 to 5, $n$ hybrids out of the 321 reference hybrids in the training set of cross-validation were randomly selected and form the reference hybrid subsets, together with the lines respectively in scenario 2 to 5 to constitute the training sets with different sizes, where $n$ ranged from 5% to 100%. In scenario 1, the parents of sampling reference hybrids formed the training set. The results were given in Fig 5. For all methods, the prediction accuracies in scenario 1 to 5 were all growing with the sampling rate of reference hybrids $n$ increased from 5% to 100% and reached a plateau when $n$ became 40%. The specific sample size was shown in S8 Table. The increasing trend of prediction accuracies in all scenarios were similar for different prediction methods, except for scenario 2 in which the two Bayesian methods displayed no apparent improvement when $n$ increased from 5% to 10%. The prediction accuracies of two BLUP methods were significantly higher than those of the two Bayesian approaches in scenario 2 when $n < 20\%$, but that was similar when $n \geq 20\%$. Overall, the BLUP methods performed superiorly to the Bayesian approaches when the training set is small ($n < 20\%$) and the male parents of the test hybrids were not used, i.e., scenarios 1–3.

### 3.6 GBLUP separately fitting trait-associated and -unassociated markers

Since genome wide markers can be used to identify markers associated with trait, i.e., TA-SNPs, via GWAS, it might be better if TA-SNPs and trait-unassociated markers were fitted separately in GS. The TA-SNPs can be separately fitted either as fixed effect or as random effect (see Method for details). We first performed GWAS analysis using all parental lines and selected the TA-SNPs with P-value incurring the overall highest prediction accuracies in MAS, i.e., $< 0.005$, for prediction. The result was shown in Fig 6A, where the prediction accuracies of scenario 1 to 5 with the TA-SNPs incorporated in GBLUP as fixed effect were 0.566, 0.639, 0.613, 0.616, 0.655, respectively. In contrast, the prediction accuracies of scenario 1 to 5 with the TA-SNPs incorporated in GBLUP as random effect were 0.573, 0.665, 0.628, 0.625 and 0.666, respectively, which were all significantly higher than that with TA-SNPs used as fixed effect. Among all scenarios with the TA-SNPs incorporated in GBLUP either as fixed or random effect, the prediction accuracy of scenario 5 was the highest (0.655 and 0.666, respectively).

Next, we conducted GWAS analysis using all parental lines and hybrids and selected the TA-SNPs also with the P-value $< 0.005$ for prediction, which was shown in Fig 6B, where the prediction accuracies of scenario 1 to 5 with the TA-SNPs incorporated in GBLUP as fixed effect were 0.622, 0.705, 0.678, 0.671, 0.691, respectively. In contrast, the prediction accuracies

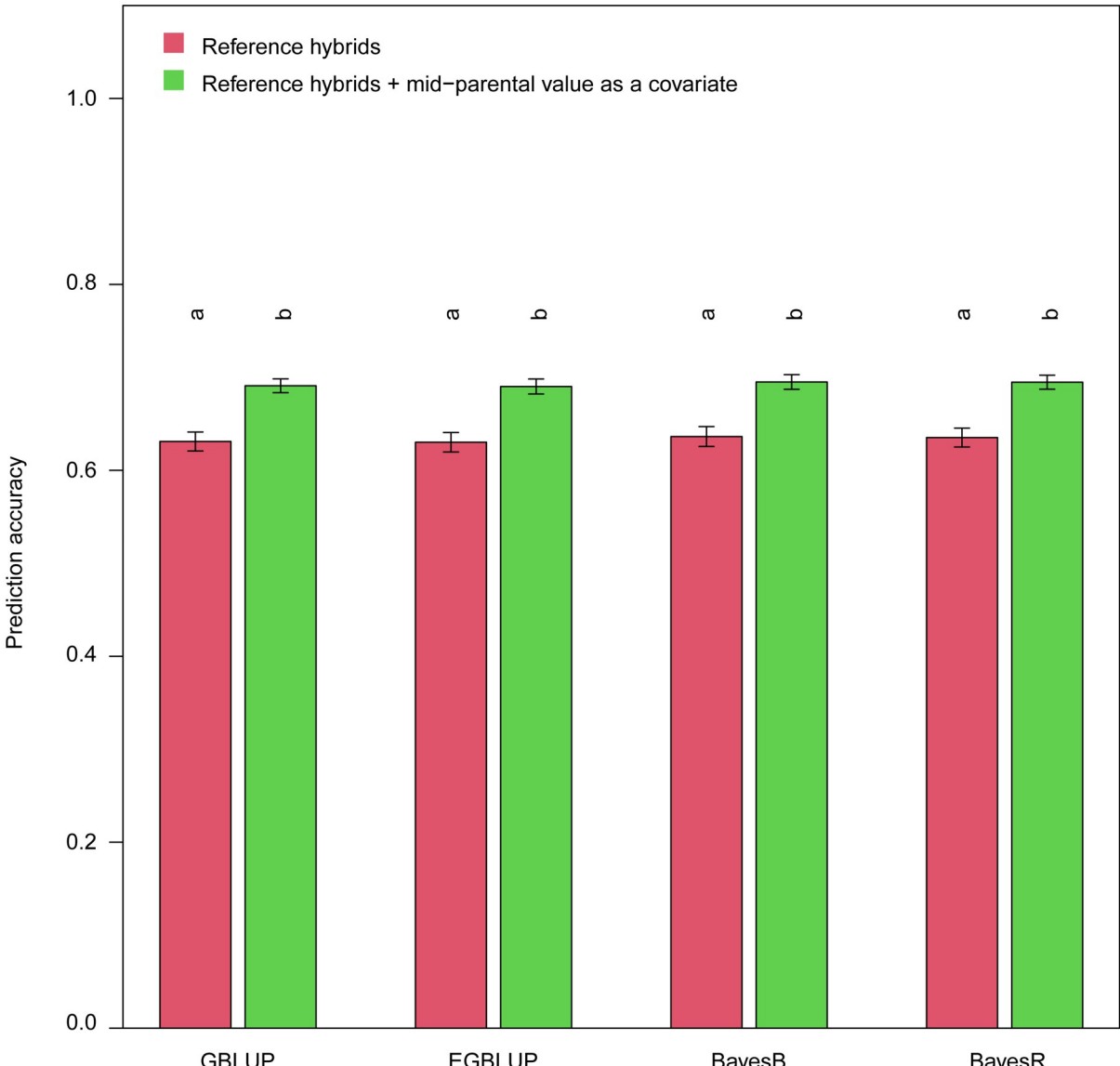

**Fig 4. Genomic prediction accuracies of mesocotyl length using mid-parental value as a covariate in the genomic prediction models.**
Different letters above the bars indicated the genomic prediction accuracies after Fisher's z-transformation were significantly different
($p < 0.05$, $t$-test) between the two scenarios.

of scenario 1 to 5 with the TA-SNPs incorporated in GBLUP as random effect were 0.616, 0.703, 0.669, 0.664 and 0.687, respectively. Interestingly, the prediction accuracies of all scenarios with the TA-SNPs incorporated as fixed effect were higher than that with TA-SNPs incorporated as random effect, especially for scenario 1, 3 and 4, where significant differences were found between them. Among all scenarios with the TA-SNPs incorporated in GBLUP either as fixed or random effect, the prediction accuracy of scenario 2 was the highest (0.705 and 0.703, respectively).

We also compared the prediction accuracies of different scenarios with TA-SNPs incorporated in GBLUP as fixed effect or random effect and those with undifferentiated using all SNPs in GBLUP. When the TA-SNPs were from GWAS based on all lines, some significant increases

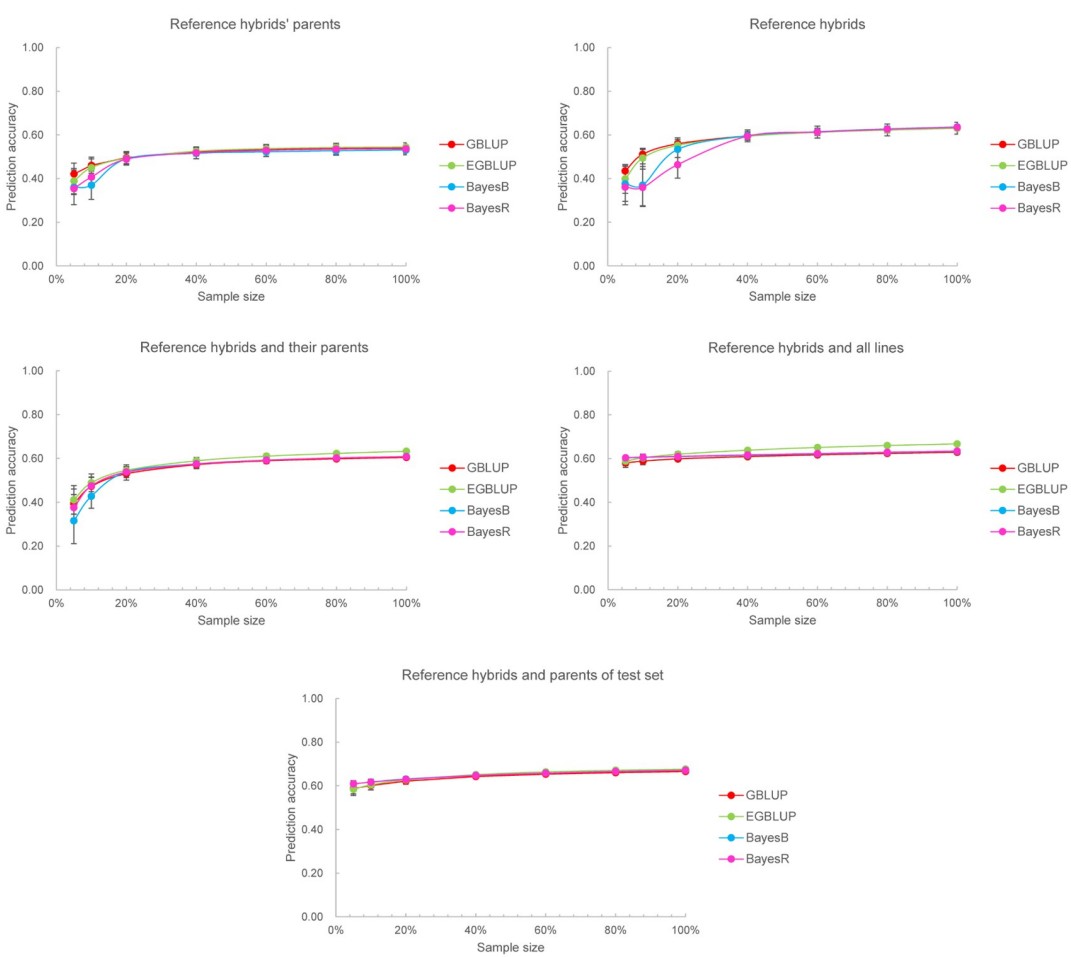

**Fig 5. Genomic prediction accuracies of mesocotyl length using four prediction models based on different training set composition scenarios and sizes.** The training set size varied resulted from the alteration of the number of reference hybrids involved in the training set.

were found between the prediction accuracies of scenario 1 to 3 with TA-SNPs incorporated in GBLUP as fixed effect and those with undifferentiated use of all SNPs, and some significant reductions were found between the prediction accuracies of scenario 4 and 5 with TA-SNPs incorporated in GBLUP as fixed effect and those with undifferentiated use of all SNPs (Fig 6A).

Meanwhile, the prediction accuracies of scenario 1 to 3 with TA-SNPs incorporated in GBLUP as random effect were significantly higher than those with undifferentiated use of all SNPs, and significant reduction was found between the prediction accuracy of scenario 4 with TA-SNPs incorporated in GBLUP as random effect and that with undifferentiated use of all SNPs (Fig 6A). No difference was found between the prediction accuracy of scenario 5 with TA-SNPs incorporated in GBLUP as random effect and that with undifferentiated use of all SNPs (Fig 6A). In contrast, when the TA-SNPs were GWAS analyzed based on all lines and hybrids, no matter incorporating TA-SNPs as fixed effect or as random effect in GBLUP, the prediction accuracies of scenario 1 to 5 were significantly higher than those with undifferentiated use of all SNPs (Fig 6B).

In summary, the best choice for modeling was accommodating the TA-SNPs from GWAS based on all lines and hybrids as fixed effect in GBLUP under scenario 2.

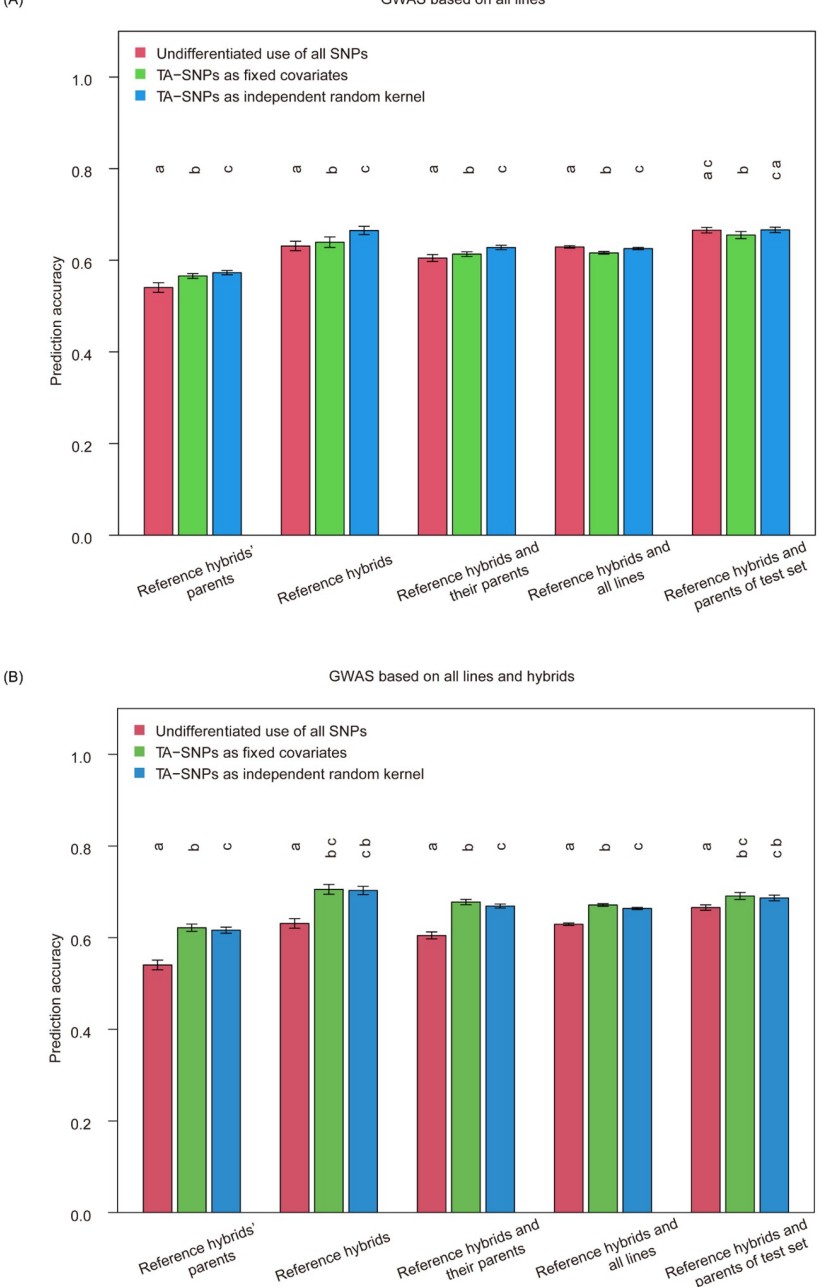

**Fig 6. Genomic prediction accuracies of mesocotyl length based on different training set composition scenarios using GBLUP by partitioning SNP markers into trait-associated and -unassociated sets using genome-wide association study (GWAS) based on all lines (A) and all lines and hybrids (B).** The trait-associated SNPs (TA-SNPs) were respectively used as fixed covariates (fixed effect) and independent random kernel (random effect) in the GBLUP model. Undifferentiated use of SNPs was using all SNPs in the GBLUP model. Different letters above the bars indicated the genomic prediction accuracies achieved by varying treatments of SNPs in the model were significantly different ($p < 0.05$, $t$-test) after a Fisher's z-transformation.

## 4. Discussion

This study demonstrated the potential of GS in breeding hybrid rice varieties suited for direct seeding capitalizing on an indicative trait mesocotyl length. We based on 401 hybrid

combinations from test-crossing 401 sequenced rice varieties from Southeast Asia, Guangdong and South Asia with a sequenced variety Taifeng A to underpin the prediction of mesocotyl length in hybrid rice. The inbred lines used as male parents are the ancestral parents of many elite varieties, and genetically diverse. Taifeng A is a female sterile line with excellent agronomic characters and is widely used in developing hybrid varieties. Therefore, the results from our study have a high practical value.

### 4.1 Relatedness driving the prediction accuracy in MAS

Previous studies have demonstrated that the relatedness between the training and test sets could impact the prediction accuracy in MAS [31,32]. This finding is validated in our study in rice. The training set composition scenarios 4 and 5 included the parents of test hybrids and the prediction accuracies in these two scenarios were remarkably higher than those in other scenarios without the parents of test hybrids especially when the threshold of P value to determine the significant SNPs was relatively liberal which may incur redundancy of predictors and impede the predictability of MAS model (Fig 2). The relatively high relatedness between the training and test sets in scenarios 4 and 5 could compensate for the nuisance as compared to other scenarios. Since the relatedness is described by SNP genotypes and a reliable estimation of relatedness requires a certain number of markers, when the P value threshold was strict and only a few significant SNPs were available, the impact of relatedness is negligible and the prediction accuracies were driven by the number of available significant SNPs (Fig 2).

The impact of relatedness could also be inspected in scenarios 1–3 in which the prediction accuracies in scenario 2 were conspicuously higher than those in scenarios 1 and 3 when the P value thresholds were liberal ($\geq 0.0025$) resulting in a comparable number of available significant SNPs in the three scenarios. Theoretically, the relatedness between the training and test sets in scenario 2 is higher than that in scenarios 1 and 3 because the male parents of the reference hybrids were genetically distant to the test hybrids thus including them in the training set would impair the relatedness.

### 4.2 GS is superior to MAS in rice mesocotyl length prediction

In the absence of genotypic data, using the mid-parental values can realize the prediction of hybrids performances. We used the mid-parental value of the hybrids as a reference to the genomics-enabled predictions. Interestingly, the mid-parental prediction accuracy was overall comparable to that of MAS but significantly lower than that of GS, indicating that GS is an efficient genomics-enabled approach in mesocotyl length breeding of hybrid rice.

The conspicuous advantage of scenarios 2–5 over scenario 1 could be attributed to the accommodation of dominant effects in addition to additive effects and also the relatedness exploited because the male parents of the reference hybrids are genetically distant to the test hybrids. The superiority of scenario 5 over scenario 4 substantiates the importance of relatedness (Fig 3). The advantage of EGBLUP over other genomic prediction approaches in scenarios 3 and 4 indicates using more inbred lines is helpful to capture the epistatic effects (Fig 3).

### 4.3 Incorporating parental performance as covariates improves prediction accuracy

Compared to the genomic prediction based solely on genomics data, including parental phenotypes in the model could significantly improve the prediction accuracy [13]. Our study underpinned this finding (Fig 4). What is worth to notice is the magnitude of reference information contained in the training set composition scenario 4 was identical to that in the scenario using mid-parental values as phenomics data in the model, however, the prediction

accuracies in the former scenario were significantly lower than those in the latter scenario (Figs 3 and 4), which indicates using mid-parental values as phenomics data in GS is a more efficient way to exploit the parental information. Xu et al. [13] mentioned using the parental phenotypes as a covariate (predictor) in the model might intrinsically capture environmental effects and genotype-by-environment interactions. In breeding, breeders could learn from it as the phenotypes of parental lines are often available prior to the crossing, therefore, no additional spending is needed.

### 4.4 Separately modelling the trait-associated and -unassociated markers significantly improved the genomic prediction accuracy

Previous studies have demonstrated that the predictability would be significantly enhanced by integrating the associated markers into the model in GS [33–35]. Here, we found similar results. The prediction accuracies for all scenarios with associated markers, which were identified from GWAS analysis using all parental lines and hybrids, either as fixed effect or random effect, were significantly higher than those undiscriminatingly using all the SNPs (Fig 6B). As compared, the advantage of distinguishingly using the SNPs in genomic prediction reduced when the GWAS was implemented only using the lines (Fig 6A). This could be explained by that expanding the population for GWAS would enhance the power of identifying trait-associated makers thereupon enhancing genomic predictability.

Comparing the effectiveness of using TA-SNPs as fixed and random effect, when GWAS was conducted using all the parental lines and hybrids, fitting the TA-SNPs as fixed effect in the genomic prediction model was overall better than that as random effects. However, the precedence was reversed when GWAS was implemented only based on the parental lines (Fig 6). This could be attributed to that GWAS using all lines and hybrids is more powerful and able to identify more reliable TA-SNPs. Because being a fixed effect in the linear model mostly would have stronger effect relative to being a random effect, a more reliable identification of trait-marker association in GWAS could underpin the fixed effect treatment. If the GWAS is not so powerful, a more conservative usage of treating the trait-associated makers as random effects would be more proper.

Overall, prior to implementing GS, using GWAS to identify trait-associated markers and discriminatingly modelling the trait-associated and -unassociated markers in GS models is suggested.

## 5. Conclusion

Based on a population of 402 rice lines and their 401 hybrid combinations, we demonstrated that using half-sib hybrids as the training set together with the mid-parental phenotypic values of all hybrids fitted as a covariate in the genomic models could achieve an optimal prediction of mesocotyl length, which is indicative of rice direct seeding ease. Including approximately 60 hybrids (20% of total hybrids) in the training set is able to obtain a comparable prediction accuracy to using all hybrids. Dividing the SNPs into trait-associated and -unassociated groups using GWAS based on the entire population could further improve the prediction accuracy. In practice, we suggest to firstly implement GWAS to differentiate the trait-associated and -unassociated markers based on all observations, and then using the phenotyped hybrids as training set to predict the untested hybrids with their parents' phenotypes fitted as a covariate in the GP models separately accommodating the trait-associated and -unassociated markers.

## Supporting information

**S1 Fig. Illustration of different training set composition scenarios.**
(TIF)

**S2 Fig. Heterosis analysis of 401 hybrids.**
(TIF)

**S1 Table. The information of 402 rice accessions.**
(DOCX)

**S2 Table. The genotypic data of the 402 lines.**
(XLSX)

**S3 Table. The best unbiased estimated values (BLUE) of mesocotyl length of 402 rice accessions and 401 hybrids.**
(DOCX)

**S4 Table. Heterosis analysis of 401 hybrids.** Mid-parent value is the average best linear unbiased estimates (BLUE) of mesocotyl length of parents of the hybrids. d-value is the difference of BLUE between hybrid and its mid-parent value. a-value is absolute value of the difference of BLUE between parents. Hp is d-value / a-value. High parent heterosis (HPH) represents the Hp of hybrid was over 1. Mid-parent heterosis (MPH) represents the Hp of hybrid was over 0 and below 1 (including 1). Low-parent heterosis (LPH) represents the Hp of hybrid was over -1 (including -1) and below 0. Hybrid inferiority (HI) represents the Hp of hybrid was below -1.
(DOCX)

**S5 Table. The prediction accuracies of scenario 1–5 in MAS.** The asterisks indicate the Fisher's z-transformed prediction accuracies in scenarios 1–5 were significantly higher ($p < 0.05$, $t$-test) than the Fisher's z-transformed mid-parental value prediction accuracy.
(DOCX)

**S6 Table. Two-Way ANOVA in MAS prediction accuracies.** Scenario and P value are the two factors. Scenario:P value represents the interaction effect between scenario and P value. Df represents degree of freedom. SS represents sum of squares. MS represents mean squares. F value is MS / $MS_{Error}$. P (F) is the P value of $F$-test. All prediction accuracies were Fisher's z-transformed.
(DOCX)

**S7 Table. Two-Way ANOVA in GS prediction accuracies.** Scenario and prediction model are the two factors. Scenario:prediction model represents the interaction effect between scenario and prediction model. Df represents degree of freedom. SS represents sum of squares. MS represents mean squares. F value is MS / $MS_{Error}$. P (F) is the P value of $F$-test. All prediction accuracies were Fisher's z-transformed.
(DOCX)

**S8 Table. The specific sample size of training set in all scenarios.**
(DOCX)

## Acknowledgments

The authors thank Sanwen Huang, Agricultural Genomics Institute in Shenzhen, Chinese Academy of Agricultural Sciences, Shenzhen, China for providing all the necessary test facilities.

## Author Contributions

**Conceptualization:** Sang He, Guoyou Ye.

**Data curation:** Jindong Liu, Sang He.

**Formal analysis:** Liang Chen, Sang He.

**Funding acquisition:** Guoyou Ye.

**Investigation:** Liang Chen.

**Methodology:** Sang He, Guoyou Ye.

**Project administration:** Liyong Cao, Guoyou Ye.

**Supervision:** Liyong Cao, Guoyou Ye.

**Validation:** Liang Chen.

**Writing – original draft:** Liang Chen.

**Writing – review & editing:** Sang He, Guoyou Ye.

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
