## [Decision Letter · Decision Letter 0]

9 Feb 2023

PONE-D-22-25635Genomic prediction of rice mesocotyl length indicative of directing seeding suitability using a half-sib hybrid populationPLOS ONE

Dear Dr. He,

Thank you for submitting your manuscript to PLOS ONE. After careful consideration, we feel that it has merit but does not fully meet PLOS ONE’s publication criteria as it currently stands. Therefore, we invite you to submit a revised version of the manuscript that addresses the points raised during the review process.

We look forward to receiving your revised manuscript.

Kind regards,

Muhammad Abdul Rehman Rashid, PhD

Academic Editor

PLOS ONE

and https://journals.plos.org/plosone/s/file?id=ba62/PLOSOne_formatting_sample_title_authors_affiliations.pdf.

“This work was supported by National Key R&D Program of China (2020YFE0202300)，and the Agricultural Science and Technology Innovation Program (ASTIP).”

“The authors acknowledge the support of Director Sanwen Huang, Agricultural Genomics Institute in Shenzhen, Chinese Academy of Agricultural Sciences, Shenzhen, China for providing all the necessary facilities including the funding for conducting the experiment.

“This work was supported by National Key R&D Program of China (2020YFE0202300)，and the Agricultural Science and Technology Innovation Program (ASTIP).”

Reviewers' comments:

Reviewer's Responses to Questions

**Comments to the Author**

1. Is the manuscript technically sound, and do the data support the conclusions?

Reviewer #1: Partly

Reviewer #2: Yes

Reviewer #3: Partly

2. Has the statistical analysis been performed appropriately and rigorously? 

Reviewer #1: Yes

Reviewer #2: Yes

Reviewer #3: I Don't Know

3. Have the authors made all data underlying the findings in their manuscript fully available?

Reviewer #1: Yes

Reviewer #2: No

Reviewer #3: Yes

4. Is the manuscript presented in an intelligible fashion and written in standard English?

Reviewer #1: Yes

Reviewer #2: No

Reviewer #3: No

5. Review Comments to the Author

Reviewer #1: The authors have presented a meaningful study to predict rice mesocotyl length, which is an indicative trait associated with emergence rate, early vigor, and lodging tolerance. The authors have used half-sib hybrids to investigate the accuracy of several prediction models. The authors have demonstrated that a two-step linear mixed model with parental information can be advantages in prediction.

But before acceptance of publication, there are still some parts unclear to me.

First, why filter out the SNPs with heterozygotes rate more than 10%? Including heterozygotes might tell additive effects from dominant effects.

Second, why ever examining using TA-SNPs as random effects? In mixed linear model, the expectation of random effects should be 0. But the contribution from TA-SNPs would be strong to predict the dependence variable.

Third, what is the protocol to select p-value threshold of significant SNPs? seems that it is arbitrary.

The caption of Figure 3 is hard to interpret. What does the different letters above the bars mean? How do the letters show "the genomic prediction accuracies of varied scenarios were significantly different"? How to tell from the letters what is significant?

Overall, the manuscript is well-written and promising. The methodology is clear for reproducibility.

Reviewer #2: Its a problem with many papers pertaining to prediction models, that instead of explaining the tangible working principle of their prediction methods BLUP etc, They tend to write matrix equations on paper too much. look at the end its a paper intended to improvise plant breeding process, you should focus on how actually SNPs data is collected and how it is applied to phenotypic data (comparative application) and how SNP variances are related to trait variances. Your paper focuses on applications of different strategies to predict g values etc. but it doesnt tell how these prediction strategies are working actually. In nutshell there is a lot of ambiguity pertaining to the process of mathematical application in the experiment. As a breeder this paper doesnt seem helpful and it takes a learner further away from modern available education. The paper should not focus on names of models and softwares involved but on the tangible explainations of processes involved.

Reviewer #3: This manuscript has scientific merit and could contribute to the literature, but unfortunately in its present form it is full of grammatical errors. These errors make the manuscript difficult to read and effectively evaluate.

Major concerns center around the clarity of the methods section. The experimental design is poorly described, and it is not clear how parental information was incorporated into the models for prediction scenarios including parental information. I assume the parents’ genotypes were included in the training of the Bayesian approaches and in the calculation of the genomic relationship matrices for the GBLUP models; however, I did not see this explicitly stated and there were comments about the collinearity of the additive and dominance relationship matrices that make me question whether this was the case.

Specific comments related to grammar.

Initially I highlighted issues with grammar and typos but gave up once I got to the methods section given the many errors. Below are the issues I highlighted before abandoning the effort:

Typo line 53: normally have short mesocotyl (≤1.0 cm) [6]. Thereby, it is of crucial to breed long

53 mesocotyl hybrid varieties for direct seeding. – need to reword

Line 74: Thereby, it is recommended to incorporate non-additive effects despite the additive effects are determinant [12]. – need to reword.

Line 87: metabolome data and eight GS methods, founding that the GBLUP approach integrating genomics and metabolome data performed overall the best. – Should be finding?

Line 91: traits of hybrid rice, but no study ever reported the potential of

GS on mesocotyl length in hybrid rice, which is indicative to direct seeding – need to reword

Line 95: We experimented several genomic prediction scenarios combining with mid-parental value prediction, marker-assisted selection (MAS) and genome-wide association study (GWAS). – Should be examined several … ?

Line 98: with highest prediction accuracy of mesocotyl length whereby disclose the potential of using genomic selection to improve hybrid rice direct seeding efficiency. – need to reword

Line 144: In consequence, 196,640 high quality SNPs retained. – Should be As a result, ...?

line 178: validation scenarios 1-5 (detailedly introduced in section 2.8) – details can be found in section 2.8. - detailedly is rarely, if ever, used in English.

Line 210: could be uniformed as = + + – do you mean represented as?

Specific comments related to methods:

You never justify why you are using mesocotyl length as a proxy for improving emergence as opposed to measuring emergence directly. Is it more cost effect? Is it more heritable? Too challenging to generate enough seed?

Experimental design – There needs to be a better description of the experimental design. Was there no replication for any of the hybrids? What is meant by partially randomized (is this in reference to parental lines being planted next to hybrids)? Was there some type of blocking? At one point the manuscript mentions corrected phenotypes, which implies some sort of correction based on the experimental design, but none of the models have any experimental factors included.

However, as only one female was used in our study, deductively, the additive genomic profiles of the hybrids were completely collinear with their dominant genomic profiles.

- The dominance relationship of the hybrids would be colinear with the additive relationship of the male inbreds. The additive relationship of the hybrids would be different (higher) as they all share a common female parent. You should be clear in the methods how you calculated the relationship matrix and modify this statement accordingly.

The iteration times of all models were uniformly set to 30,000 and first 5,000 times were set as burn-in.

- Did you confirm burn-in and good mixing of the Markov chains? No thinning was needed?

For cross-validation scenarios 1 and 5

- For GBLUP were the parents included in the genomic relationship matrix? If that is the case, there would be a large difference in the dominance and additive relationship matrices.

Line 259 …the corresponding adjusted phenotypic values was calculated to measure the genomic prediction accuracy.

- How were the phenotypic values adjusted?

In the mid-parental value prediction trial, as the five hybrid test sets in each repeat of cross-validation were collectively the total hybrid population disregarding to the randomness of sampling, thus there was just one prediction accuracy value in the scenario using the mid-parental values as a covariate.

- This statement is confusing. Perhaps some type of figure to illustrate the various cross-validations schemes would help make this clearer.

- It is confusing how parental information is included for prediction in the models.

Line 296 The additive effect variance component was 1.22 for parent lines and

1.7 for hybrids. The heritability estimate was 0.8 for parent lines and 0.58 for hybrids.

The genetic diversity of the parental lines was overall high, as indicated by the wider

range of genetic similarities between parental lines

- Which model was used to calculate the additive variance components?

6. PLOS authors have the option to publish the peer review history of their article (what does this mean?). If published, this will include your full peer review and any attached files.

Reviewer #1: **Yes: **Weihao Ge

Reviewer #2: No

Reviewer #3: No

---

## [Author Response · Author response to Decision Letter 0]

14 Mar 2023

Dear editor:

We have revised the manuscript according to the constructive comments from the reviewers and you. The point-by-point responses have been given to each comment. All revisions in the manuscript were marked in red text. Thanks for all the comments.

Reviewers’ comments:

Reviewer #1:

The authors have presented a meaningful study to predict rice mesocotyl length, which is an indicative trait associated with emergence rate, early vigor, and lodging tolerance. The authors have used half-sib hybrids to investigate the accuracy of several prediction models. The authors have demonstrated that a two-step linear mixed model with parental information can be advantages in prediction. But before acceptance of publication, there are still some parts unclear to me.

1. Why filter out the SNPs with heterozygotes rate more than 10%? Including heterozygotes might tell additive effects from dominant effects.

Response: Thanks for the comment. We agreed that including more heterozygotes can tell additive effects from dominant effects. However, the SNP genotypes of hybrids were deduced from the genotypes of their parental lines. Heterozygotes would incur uncertainty in the deduction of genotypes of hybrids. The ideal situation is all the SNPs are completely genotyped and there is no missing value. But in reality it is often not the case. Therefore, we could alternatively wipe out the heterozygotes by setting them as missing values and then impute them. If the heterozygotes rate is high, arbitrarily setting heterozygotes as missing values is too manipulative and unreasonable. Therefore, we excluded the SNPs with heterozygotes rate more than 10% to keep a relatively low heterozygotes rate.

2. Why ever examining using TA-SNPs as random effects? In mixed linear model, the expectation of random effects should be 0. But the contribution from TA-SNPs would be strong to predict the dependence variable.

Response: Thanks for the remark. Sometimes if the trait of interest is complex, there would be a large number of TA-SNPs (also dependent on the significance threshold of P value) present. Fitting them as fixed effect could also raise the “large p, small n” problem and incur convergence issue of linear models. Therefore, using them as random effects is more proper. There are also some studies ever examining the TA-SNPs as random effects, for instance, the BayesRC approach proposed by MacLeod et al. (2016)

(MacLeod I M, Bowman P J, Vander Jagt C J, et al. Exploiting biological priors and sequence variants enhances QTL discovery and genomic prediction of complex traits. BMC genomics, 2016, 17: 1-21.).

3. What is the protocol to select p-value threshold of significant SNPs? seems that it is arbitrary. The caption of Figure 3 is hard to interpret. What does the different letters above the bars mean? How do the letters show "the genomic prediction accuracies of varied scenarios were significantly different"? How to tell from the letters what is significant?

Response: Thanks for the comment. 1) The p-value threshold was chosen based on the number of significant SNPs available. If the threshold is too strict, there was no significant SNP available in some scenarios and it will cause imbalance between the scenarios. The threshold is also not needed to be too liberal because the number of effective significant SNPs would not markedly increase until the threshold rose to 0.01.

2) The letters in Figure 3 represent that within the same model algorithm, the prediction accuracies in different scenarios were significantly different or not at the level of p < 0.05. The letters a-e refer to the scenarios 1-5 (from left to right). If two scenarios share a letter, it means the prediction accuracies in these two scenarios were not significantly different. Otherwise, if two scenarios have no common letter, it means the prediction accuracies in these two scenarios were significantly different. We have augmented in the Figure 3 caption. This can be seen in line 413-415.

Overall, the manuscript is well-written and promising. The methodology is clear for reproducibility.

Reviewer #2:

It’s a problem with many papers pertaining to prediction models, that instead of explaining the tangible working principle of their prediction methods BLUP etc, They tend to write matrix equations on paper too much. look at the end its a paper intended to improvise plant breeding process, you should focus on how actually SNPs data is collected and how it is applied to phenotypic data (comparative application) and how SNP variances are related to trait variances. Your paper focuses on applications of different strategies to predict g values etc., but it doesn’t tell how these prediction strategies are working actually. In nutshell, there is a lot of ambiguity pertaining to the process of mathematical application in the experiment. As a breeder this paper doesn’t seem helpful and it takes a learner further away from modern available education. The paper should not focus on names of models and softwares involved but on the tangible explainations of processes involved. 

Response: Thanks for the comment. Our intention is just to provide an optimal way to predict hybrids’ performance capitalizing on genomic data. It is not a study systematically directing the application of genomic prediction in breeding. We agreed the application is important and we added one conclusive sentence in the conclusion section in line 610-615 with red font. We also enriched the explanation of different scenarios to make the scenarios clearer and more tangible. Please see the red text in the Materials and methods section.

Reviewer #3:

This manuscript has scientific merit and could contribute to the literature, but unfortunately in its present form it is full of grammatical errors. These errors make the manuscript difficult to read and effectively evaluate.

1. Major concerns center around the clarity of the methods section. The experimental design is poorly described, and it is not clear how parental information was incorporated into the models for prediction scenarios including parental information. I assume the parents’ genotypes were included in the training of the Bayesian approaches and in the calculation of the genomic relationship matrices for the GBLUP models; however, I did not see this explicitly stated and there were comments about the collinearity of the additive and dominance relationship matrices that make me question whether this was the case. 

Response: Thanks for the remark. 1) the introduction of experimental design has been enriched in line 110-113 with red text. 2) What you assumed is right, the parents’ genotypes together with hybrids’ genotypes were all included in one genomic relationship matrix for scenario 1 and 3-5. But for the scenario incorporated mid-parental value, we took mid-parental value as a covariate incorperated into the model. We have elaborated the models to clarify the use of parental information. Please see the red text in Materials and methods section 2.7 from line 203-254.

2. Specific comments related to grammar.

Initially I highlighted issues with grammar and typos but gave up once I got to the methods section given the many errors. Below are the issues I highlighted before abandoning the effort:

2.1 Typo line 53: normally have short mesocotyl (≤1.0 cm) [6]. Thereby, it is of crucial to breed long 53 mesocotyl hybrid varieties for direct seeding. – need to reword.

Response: We have revised this sentence. Please see line 53-54 with red text.

2.2 Line 74: Thereby, it is recommended to incorporate non-additive effects despite the additive effects are determinant [12]. – need to reword.

Response: We have reworded this sentence. Please see line 75-76 with red text.

2.3 Line 87: metabolome data and eight GS methods, founding that the GBLUP approach integrating genomics and metabolome data performed overall the best. – Should be finding?

Response: Yes, we have revised and please see line 88-90 with red text.

2.4 Line 91: traits of hybrid rice, but no study ever reported the potential of GS on mesocotyl length in hybrid rice, which is indicative to direct seeding – need to reword.

Response: This sentence has been revised. Please see line 92-93 with red text.

2.5 Line 95: We experimented several genomic prediction scenarios combining with mid-parental value prediction, marker-assisted selection (MAS) and genome-wide association study (GWAS). – Should be examined several … ?

Response: We have revised this sentence. Please see line 96-98 with red text.

2.6 Line 98: with highest prediction accuracy of mesocotyl length whereby disclose the potential of using genomic selection to improve hybrid rice direct seeding efficiency. – need to reword.

Response: We have reworded. This can be seen in line 99-101 with red text.

2.7 Line 144: In consequence, 196,640 high quality SNPs retained. – Should be As a result, ...?

Response: We have revised. Please see line 152-153 with red text.

2.8 Line 178: validation scenarios 1-5 (detailedly introduced in section 2.8) – details can be found in section 2.8. - detailedly is rarely, if ever, used in English.

Response: We have revised. Please see line 188 with red text.

2.9 Line 210: could be uniformed as = + + – do you mean represented as?

Response: We have revised. Please see line 228 with red text.

3. Specific comments related to methods:

3.1 You never justify why you are using mesocotyl length as a proxy for improving emergence as opposed to measuring emergence directly. Is it more cost effect? Is it more heritable? Too challenging to generate enough seed?

Response: Thanks for the comment. Mesocotyl length is significantly positively correlated with the emergence rate and in practice commonly used to evaluate the effect of direct seeding. Emergence is also important for direct seeding, however, measuring emergence requires more seeds. It is not cost-effective and too challenging to generate enough hybrid seeds. Therefore, we chose the length of mesocotyl as a proxy of emergence.

3.2 Experimental design – There needs to be a better description of the experimental design. Was there no replication for any of the hybrids? What is meant by partially randomized (is this in reference to parental lines being planted next to hybrids)? Was there some type of blocking? At one point the manuscript mentions corrected phenotypes, which implies some sort of correction based on the experimental design, but none of the models have any experimental factors included.

Response: Thanks for the comment. We have added the details of experimental design including the number of replicates for the test of mesocotyl length measurement to the Materials and Methods section. This can be seen in line 110-113 with red text. 1) We have set three replicates for hybrids. 2) In order to minimize the impact of environment on the phenotypic performances of parents and their hybrid, each hybrid was planted next to its male parent. The spot of each hybrid planting totally depends on its male parent, therefore, we called it partially randomized. But considering “partially randomized” will incur confusion, we have removed it in the manuscript. 3) We have elaborated the phenotype correction process in line 124-130 with red text. Please check it.

3.3 However, as only one female was used in our study, deductively, the additive genomic profiles of the hybrids were completely collinear with their dominant genomic profiles. - The dominance relationship of the hybrids would be colinear with the additive relationship of the male inbreds. The additive relationship of the hybrids would be different (higher) as they all share a common female parent. You should be clear in the methods how you calculated the relationship matrix and modify this statement accordingly.

Response: Thanks for the comment. Let us exemplify it. For a given locus, assuming the genotypic profiles of five male parents are (0 2 0 0 2). As all hybrids share one female, the genotypic profiles of the five hybrids should be (0 1 0 0 1) if the genotypic profile of female is 0, and (1 2 1 1 2) if the genotypic profile of female is 2. The corresponding dominant genotypic profile should (0 1 0 0 1) and (1 0 1 1 0) because only heterozygotes have dominant effect. It is clear that (0 1 0 0 1) is colinear with (0 1 0 0 1), and (1 2 1 1 2) is colinear with (1 0 1 1 0). The additive and dominant relationship matrices could be uniformly described as WWT/c where W is either additive or dominant genotypic matrix, c is a constant. The standardization of W applied in many studies is just a linear transformation of W, which will not change the rank of W. Theoretically, the rank of WWT is same as W. So, if the additive profile matrix is colinear with the dominant profile matrix, the additive relationship matrix must be colinear with the dominant relationship matrix.

3.4 The iteration times of all models were uniformly set to 30,000 and first 5,000 times were set as burn-in. - Did you confirm burn-in and good mixing of the Markov chains? No thinning was needed?

Response: Thanks for the remark. The thinning for BLUP models was 5 (default setting in BGLR package) and for Bayesian models was 10 (default setting in gctb software). In order to uniform the thinning, we have reset the thinning of Bayesian models to be 5 and rerun all the scenarios using Bayesian models in our study. It was shown that there is no observable difference between the results by setting the thinning being 5 and 10. We randomly selected some scenarios using GBLUP and BayesB (after resetting thinning=5) to show the convergence of estimates of mu, variance components of genotype (var_a) and residual (var_e) in the MCMC as:

As shown, there were good convergences of all the estimates for both models.

3.5 For cross-validation scenarios 1 and 5 - For GBLUP were the parents included in the genomic relationship matrix? If that is the case, there would be a large difference in the dominance and additive relationship matrices.

Response: Thanks for the comment. The parents were included in the relationship matrix in scenarios 1 and 3-5. As parents are all pure lines, there is no heterozygote in parents. So even the dominant relationship matrix was constructed, all the elements corresponding to lines per se (diagonal) or relatedness between line and hybrid (off-diagonal) are zero. The elements corresponding to hybrids per se and relatedness between hybrids are not zero but this submatrix is collinear with the additive relationship matrix, as introduced in comment 3.3. So, there is no need to construct the dominant relationship matrix.

3.6 Line 259 …the corresponding adjusted phenotypic values was calculated to measure the genomic prediction accuracy. - How were the phenotypic values adjusted?

Response: Thanks for the remark. We have elaborated the phenotype correction process in line 124-130 with red text. Please check it.

3.7 In the mid-parental value prediction trial, as the five hybrid test sets in each repeat of cross-validation were collectively the total hybrid population disregarding to the randomness of sampling, thus there was just one prediction accuracy value in the scenario using the mid-parental values as a covariate. - This statement is confusing. Perhaps some type of figure to illustrate the various cross-validations schemes would help make this clearer. - It is confusing how parental information is included for prediction in the models.

Response: Thanks for the comment. 1) We have elaborated the prediction accuracy calculation in mid-parental value prediction scenario. Please see line 290-299 with red font. One supplementary figure (S1 Fig) was also added to illustrate different cross-validation scenarios. 2) For scenario 1 and 3-5, the parents’ genotypes together with hybrids’ genotypes were all included in one genomic relationship matrix. But for the scenario incorporated mid-parental value, we took mid-parental value as a covariate incorporated into the model. We have elaborated the models to clarify the use of the parental information. Please see the red text in Materials and methods section 2.7 from line 203-254.

3.8 Line 296 The additive effect variance component was 1.22 for parent lines and 1.7 for hybrids. The heritability estimate was 0.8 for parent lines and 0.58 for hybrids. The genetic diversity of the parental lines was overall high, as indicated by the wider range of genetic similarities between parental lines - Which model was used to calculate the additive variance components?

Response: Thanks for the remark. The additive and other variance components used to calculate the heritability were from GBLUP model. We have added relevant text in line 225-227 with red font.

---

## [Editor Report · Decision Letter 1]

21 Mar 2023

Genomic prediction of rice mesocotyl length indicative of directing seeding suitability using a half-sib hybrid population

PONE-D-22-25635R1

Dear Dr. He,

We’re pleased to inform you that your manuscript has been judged scientifically suitable for publication and will be formally accepted for publication once it meets all outstanding technical requirements.

Kind regards,

Muhammad Abdul Rehman Rashid, PhD

Academic Editor

PLOS ONE
---

## [Editor Report · Acceptance letter]

27 Mar 2023

PONE-D-22-25635R1 

Genomic prediction of rice mesocotyl length indicative of directing seeding suitability using a half-sib hybrid population 

Dear Dr. He:

I'm pleased to inform you that your manuscript has been deemed suitable for publication in PLOS ONE. Congratulations! Your manuscript is now with our production department. 

Kind regards, 

on behalf of

Dr. Muhammad Abdul Rehman Rashid 

Academic Editor

PLOS ONE